# Extracutaneous Melanotic Melanoma with Nervous System Involvement in a Buffalo (*Bubalus bubalis*)

**DOI:** 10.3390/vetsci10120662

**Published:** 2023-11-21

**Authors:** José Diomedes Barbosa, Mariana Correia Oliveira, Carlos Magno Chaves Oliveira, Henrique dos Anjos Bomjardim, Tatiane Teles Albernaz Ferreira, Marcos Dutra Duarte, José Alcides Sarmento da Silveira, Natália da Silva e Silva Silveira, Camila Cordeiro Barbosa, Aluízio Otávio Almeida da Silva, Anibal Armién, Marilene de Farias Brito

**Affiliations:** 1Instituto de Medicina Veterinária, Universidade Federal do Pará (UFPA), Castanhal 68740-970, PA, Brazil; cmagno@ufpa.br (C.M.C.O.); tatyalbernaz@ufpa.br (T.T.A.F.); duartemd@ufpa.br (M.D.D.); jalcides@ufpa.br (J.A.S.d.S.); nataliasilva@ufpa.br (N.d.S.e.S.S.); camilabarbosamedvet@gmail.com (C.C.B.); otavio@ufpa.br (A.O.A.d.S.); 2Departamento de Epidemiologia e Saúde Pública (DESP), Instituto de Veterinária, Universidade Federal Rural do Rio de Janeiro (UFRRJ), Seropédica 23890-000, RJ, Brazil; mariana.correia@estacio.br (M.C.O.); mfariasbrito@uol.com.br (M.d.F.B.); 3Instituto de Estudos do Trópico Úmido (IETU), Universidade Federal do Sul e Sudeste do Pará (UNIFESSPA), Xinguara 68557-335, PA, Brazil; henriquebomjardim@unifesspa.edu.br; 4California Animal Health & Food Safety Laboratory System, University of California, Davis, CA 95616, USA; agarmien@ucdavis.edu

**Keywords:** buffalo, immunohistochemistry, Melan-A, neoplasm

## Abstract

**Simple Summary:**

Buffalo farming holds considerable importance in the state of Pará, Brazil, as it contributes to both the economy and culture. Therefore, it is important to understand the diseases affecting these animals, particularly neoplastic diseases affecting the nervous system. This study describes the occurrence of malignant melanotic melanoma in the nervous system without skin involvement and with multiorgan metastasis in a buffalo raised in the Amazon biome in Brazil.

**Abstract:**

Melanomas are tumors arising from externally uncontrolled melanocytes that produce varying amounts of melanin. In this study, we report a case of melanoma with neurological impairment without evidence of cutaneous neoplastic lesions in an adult buffalo in the state of Pará, Brazil. Clinically, the buffalo exhibited apathy, decreased mandibular tone, and occasionally an open mouth with motor incoordination, and eventually succumbed to the condition. Necropsy revealed multifocal tumor masses in the brain, pituitary gland, trigeminal ganglion, and spinal cord. The neoplastic cells showed strong positive signals for vimentin, Melan-A, PNL-2, and SOX10. The diagnosis was made via necropsy, histopathology, and positive immunostaining for Melan-A and PNL-2, which are specific markers for melanocyte identification.

## 1. Introduction

The Brazilian buffalo herd is estimated at 1,598,268 heads, and Pará has the largest buffalo herd in Brazil (644,672 heads) [1]. Buffaloes are rustic animals that adapt well to adverse climates. Morphologically, they are characterized by a high concentration of melanin in the skin and hair, a limited number of sweat glands, and sparse hair covering; however, they have difficulty dissipating heat and absorb solar radiation easily owing to their dark coloration [2]. Neoplasms are generally uncommon in buffaloes, with lymphoma being the most commonly diagnosed neoplasm. In Pará, cases of multicentric lymphoma leading to progressive weight loss, generalized lymphadenopathy, and abdominal distension have been reported in buffaloes. Unlike in cattle, there appears to be no correlation between enzootic bovine leukosis virus (EBV) and the development of generalized lymphoma in buffaloes, although some tests in buffaloes were positive for EBV. In one study, specific antibodies against VLEB were detected in 4 out of 14 buffaloes experimentally inoculated intra-abdominally [3]. In another study based on a double radial immunodiffusion technique with anti-VLEB antibodies, 10 out of 232 (4.3%) buffaloes tested positive for EBV [4].

Other neoplasms reported in buffaloes include tubular adenocarcinoma, oral hemangioma, rectal leiomyoma [5], melanoma [6,7], urothelial papilloma [8], intraductal mammary carcinoma [9], mesothelioma, bladder carcinoma, ductal carcinoma, biliary tract carcinoma, lymphoid leukemia, and thymic lymphoma [6].

Melanoma is a common malignant neoplasm of melanocytes, a type of cell derived from the neural crest that is found in the basal epidermis and hair follicles. However, melanocytic tumors occasionally originate in mucosal surfaces, meninges, and the choroidal layer of the eyeball [10,11]. In humans, the development of melanomas is associated with chromosomal mutations in melanocytes resulting from intense exposure to ultraviolet radiation from the sun as well as genetic causes [12]. Melanoma is common in humans and dogs, but rare in animals reared for meat. In Brazil, melanoma has been reported in goats [13,14], pigs [15], and rabbits [16], and is relatively common in grey-haired horses [17], and cattle [18,19]. It has also been reported in cattle in India [20,21], the United States [22], and Turkey [23].

Data on the occurrence of melanoma in buffaloes are scarce [6,7,24], and there are no reports of melanoma involving the nervous system in this species. In this case report, we describe the occurrence of malignant melanotic melanoma in the nervous system without skin involvement and with multi-organ metastasis in a buffalo raised in the Amazon biome in Brazil.

## 2. Case Presentation

A 12-year-old adult female Murrah buffalo with a good body score and body weight of 650 kg presented with apathy, moderate dehydration, dyspnea, wheezing, tachycardia, and fever. Clinical examination of the nervous system revealed decreased alertness and postural changes characterized by ataxia, as demonstrated by the abduction of the right thoracic limb, anorexia, decreased tongue and jaw muscle tone, the occasional opening of the mouth and tongue protrusion, loss of facial skin sensitivity, corneal opacity with loss of visual acuity in the left eyeball, and convergent strabismus in the right eyeball. Additionally, the buffalo demonstrated reluctance to move, had limited mobility, lost balance, and constantly fell to the floor. Occasionally, it walked in circles on the left side. When in the lateral or sternal decubitus position, it faced difficulty in standing up. Supportive treatment was administered in the form of intravenous fluid therapy with 0.9% sodium chloride solution (15 mL/kg/h) for 48 h and dipyrone (25 mg/kg) twice a day for 48 h to control fever; however, no clinical improvement was observed, and the animal died 2 days after clinical care.

Necropsy performed by professors from the Federal University of Pará (UFPA) revealed tumor masses in the liver, adrenal cortex, right kidney capsule, and prescapular and subiliac lymph nodes. Additional lesions, including multiple ulcers, were observed in the abomasum along with marked pulmonary interstitial emphysema.

Well-demarcated, rounded, and soft tumor masses with cream coloration and brown to black foci were observed that expanded, compressed, and replaced the gray and white matter in both the right and left hemispheres of the brain (Figure 1A–C). Additional tumor masses were found in the hypothalamus, right amygdaloid body, right and left cerebellar hemispheres, leptomeninges on the right side of the medulla, left medullary lateral funiculus (Figure 1D), trigeminal ganglion, and pituitary gland. No cutaneous masses or nodules were observed macroscopically at the mucocutaneous junctions on the mucous surfaces of the oral, nasal, and genital cavities of the buffalo.

For histopathological examination, the central and peripheral nervous systems and various viscera were fixed in 10% buffered formalin, routinely processed, and stained with hematoxylin–eosin in the Pathological Anatomy Sector of Federal Rural University of Rio de Janeiro UFRRJ.

Microscopic examination revealed the proliferation of neoplastic cells with a predominantly fusiform shape supported by a thin fibrovascular stroma (Figure 2A) and possessing moderately sized and acidophilic cytoplasm and rare melanin granules. The nucleus was ovoid or pleomorphic, with single or multiple nucleoli. Additionally, foci of tumor necrosis and rare mitotic structures, both typical and atypical (mitotic index of two structures in high-magnification fields, with a total area of 2.37 mm^2^), were observed. Neoplastic cells with the same morphological characteristics were observed in all encephalic structures containing the masses, and neoplastic foci were identified in the liver, adrenal glands, right kidney, and prescapular and subiliac lymph nodes.

Tumor masses excised from the brain were sent to a private laboratory as paraffin blocks for immunohistochemical analysis using antibodies against Melan-A, PNL-2, SOX10, synaptophysin, chromogranin, S-100, glial fibrillary acidic protein (GFAP), neurofilament, NSE, and cytokeratin.

Immunohistochemical analysis of the brain masses revealed neoplastic cells positive for vimentin, Melan-A, PNL-2, and SOX10 antibodies (Figure 2B,C), sparse immunostaining for synaptophysin and chromogranin, and no staining for cytokeratin, S-100, GFAP, neurofilament, or NSE (Figure 2D), as shown in Table 1.

## 3. Discussion

Melanoma in the buffalo was diagnosed through necropsy and histopathological and immunohistochemical examinations. The clinical symptoms strongly indicative of neurological impairment were supported by the presence of numerous pigmented tumor masses in the brain and spinal cord tissues.

The various neurological impairments observed during the animal’s life can be explained by the multifocal lesions identified in the brain. Clinical signs such as loss of jaw muscle tone, loss of facial skin sensitivity, decreased tongue tone, loss of visual acuity, and convergent strabismus resulted from the different pairs of nerves associated with these structures originating in the diencephalon and brainstem being compressed by the neoplasms. Additionally, the cerebellar lesions explained the postural changes and loss of balance in the animal [25].

Although the most common form of melanoma is cutaneous, neoplasms can also arise from melanocytes on mucosal surfaces, the eye, and leptomeninges [2]. Malignant melanoma metastases are found in various tissues and originate from neoplasms located on the skin [26].

In bovine species, cases of melanoma have been reported, affecting the skin [27], lymph nodes [18], oral cavity [18], heart [18,27], lung [18,19], liver [18,19,27], mammary glands [18], hooves [21], and spleen and kidneys [19].

Cases of melanoma in buffaloes have been reported in India [6], Pakistan [24], and Brazil in two albino buffaloes of the Murrah breed [7]. However, in contrast to most cases reported in cattle and buffaloes, the neoplasm in this case affected the central nervous system.

In the present report, no cutaneous masses or nodules, or even masses on the mucosal surfaces were found during the necropsy. Many masses were observed in the brain, and no tumors were observed on the skin, mucous membranes, or eyeballs, although metastases were identified in the organs, which led us to infer that it was an extracutaneous melanoma.

In humans, primary melanoma of unknown origin refers to metastatic melanoma in the lymph nodes, subcutaneous tissue, or visceral sites in the absence of a detectable primary tumor. While the mechanism of primary melanoma of unknown origin is unclear, malignant transformation is suspected to arise from ectopic melanocytes in the lymph nodes or other organs [28]. 

Nonetheless, primary melanocytic neoplasms of the central nervous system are rare in humans and can be focal or diffuse, benign or malignant, and arise from melanocytes located in the leptomeninges [29]. To the best of our knowledge, no primary melanoma of the central nervous system has been reported in non-human animals without cutaneous or mucosal involvement. In the present study, although neoplastic masses were predominantly found in the central nervous system, the primary site of the neoplasm could not be confirmed.

Melan-A and PNL2 antibodies exhibited diffuse positivity and are ideal melanocytic markers for domesticated species [30], including buffaloes [7]. Immunoreactivity with SOX10 antibody confirmed that neoplastic cells were derived from the neural crest [30]. The neoplastic cells showed sparse staining with chromogranin and synaptophysin antibodies, and the stained regions were interpreted as foci of neuroendocrine differentiation [31].

Differential diagnosis was also performed for other diseases that present with nervous clinical signs, such as rabies, *Ipomoea asarifolia* poisoning, bovine herpesvirus type 5 encephalitis, brain abscesses, listeriosis, lead poisoning, and salt poisoning [32,33,34,35,36,37]. In Brazil, rabies in herbivores has high morbidity and lethality and can affect different species on the same property. The histopathological findings are characterized by lymphocytic meningoencephalomyelitis and ganglioneuritis with focal or diffuse gliosis [38], findings that were not observed in the animal in the present study.

Poisoning by *I. asarifolia* was ruled out because the pasture where the animal was grazing was inspected during the technical visit, and no *I. asarifolia* was found. Furthermore, in cases of *I. asarifolia* poisoning, animals consume the plant during periods of drought when they are hungry [33]; however, such conditions were not observed on the property during this study.

Encephalitis due to bovine herpesvirus type 5 causes extensive areas of malacia in the cortex of the telencephalon of affected animals [34], in contrast to the findings of the present study, in which the proliferation of neoplastic cells was observed that showed a predominantly fusiform shape supported by thin fibrovascular stroma and possessing moderately sized and acidophilic cytoplasm and rare melanin granules. Furthermore, bovine herpesvirus type 5 infection is characterized by the presence of intranuclear inclusion bodies in astrocytes and neurons [34,39], which can be visualized via microscopic examination; however, these were not observed in the study animal.

Brain abscesses are characterized by slow evolution, chronic progression, and clinical signs similar to those presented by the animal in the study; however, during necropsy and histopathological examination, the macroscopic and microscopic characteristics of the lesion ruled out brain abscesses.

Listeriosis is an infectious disease caused by the Gram-positive bacterium *Listeria monocytogenes*, which occurs sporadically or in outbreaks, has low morbidity and high lethality, and is generally associated with silage feeding [40,41]. The disease was ruled out based on necropsy and histopathological findings, as listeriosis does not lead to significant macroscopic changes, and the lesions are characterized by microabscesses, areas of malacia, and perivascular cuffs on a microscopic level [35].

On the property studied, the animal was raised in an extensive breeding system and did not have access to sources of lead, such as battery waste, lubricants, paints, motor oils, herbicides, and lead-based insecticides. Furthermore, the animals in the present study did not receive mineral supplementation and had no history of water fasting or restrictions. Therefore, cases of lead and salt poisoning were also ruled out.

## 4. Conclusions

Melanoma in buffaloes is uncommon; therefore, providing a description is important because, although rare, it contributes important information to include this neoplasm in the range of potential differential diagnoses of diseases with neurological symptoms in buffaloes and motivates studies on extracutaneous melanomas in animals. 

Reports of neoplasms in buffaloes are rare in the literature. Further studies are needed to investigate the frequency and clinical, epidemiological, and etiopathogenic conditions of neoplasms in buffaloes. A few studies in slaughterhouses have shown relatively lower numbers compared with other species of production animals, particularly cattle. Another hypothesis is that neoplasms in buffaloes may be underdiagnosed in Brazil and other countries.

## Figures and Tables

**Figure 1 vetsci-10-00662-f001:**
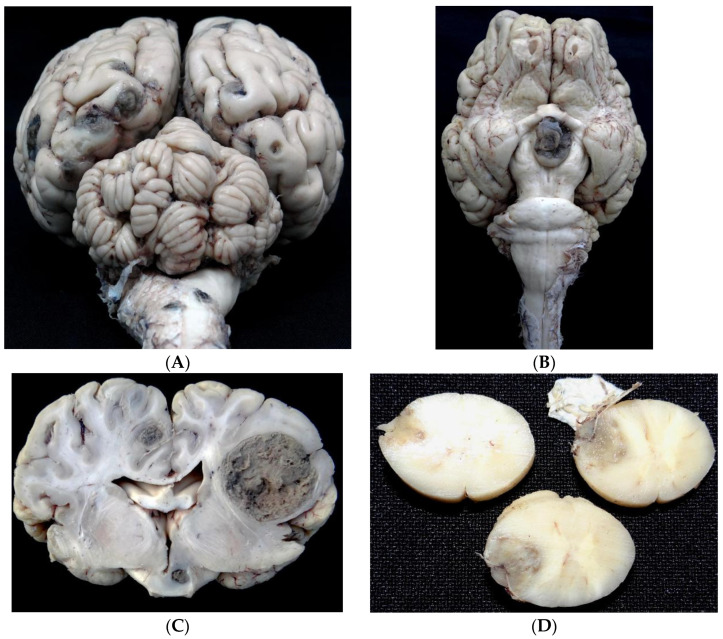
Melanotic melanoma in the nervous system of a buffalo. (**A**) Multiple tumor masses of varying sizes, randomly distributed in different areas of the cerebellum, brainstem, and spinal cord, after fixation in formalin and removal of the meninges. (**B**) Tumor mass in the hypothalamus, after fixation in formalin and removal of the meninges. (**C**) Cross-section of the brain showing multiple brown to blackish tumor masses of varying sizes involving the gray and white matter distributed randomly in the telencephalic hemispheres, diencephalon, hippocampus, and hypothalamus, after formalin fixation. (**D**) Melanotic melanoma in the nervous system of a buffalo. Submeningeal tumor masses involving the white matter of the cervical spinal cord, after formalin fixation.

**Figure 2 vetsci-10-00662-f002:**
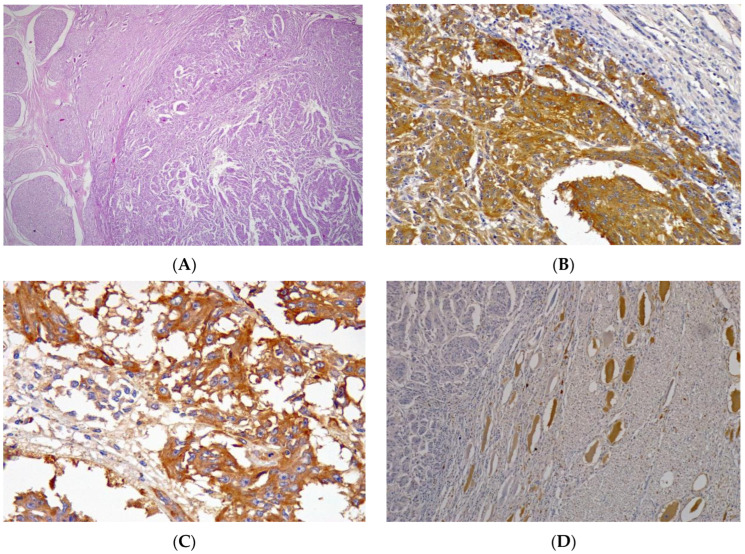
Melanotic melanoma in the nervous system of a buffalo. (**A**) Proliferation of neoplastic cells with a predominantly fusiform shape supported by thin fibrovascular stroma possessing moderately sized and acidophilic cytoplasm and rare melanin granules. The nucleus is ovoid or pleomorphic with single or multiple nucleoli. Hematoxylin–eosin staining, 4×. (**B**) Melanotic melanoma in the nervous system of a buffalo. Diffuse cytoplasmic labeling for the Melan-A antibody in neoplastic cells, 10×. (**C**) Melanotic melanoma in the nervous system of a buffalo. Diffuse cytoplasmic labeling for the PNL-2 antibody in neoplastic cells, 10×. (**D**) Melanotic melanoma in the nervous system of a buffalo. Absence of labeling for NSE antibody in neoplastic cells, 4×.

**Table 1 vetsci-10-00662-t001:** List of antibodies and results of melanoma immunohistochemistry in a 12-year-old adult Murrah buffalo. Castanhal, PA, Brazil.

Anticorpos		Clone	Results
Vimentin	Intermediate filaments of mesenchymal cells	V9	Positive
Melan-A	Melanoma antigen	A103	Positive
PNL2	Melanoma antigen	PNL-2	Positive
CK Pan	Intermediate filaments of epithelial cells	AE1AE3	Negative
Neurofilament	Marker of neurons and neuronal tumors	2F11	Negative
NSE	Neuron-specific enolase	BBS/NC/VI-H14	Negative
GFAP	Glial fibrillary acidic protein	Polyclonal	Negative
SOX10	Marker of neural crest tumors, melanomas, and schwannomas	BC34	Positive
Chromogranin	Neuroendocrine cell marker	Polyclonal	Sparsely positive
Synaptophysin	Neuroendocrine cell marker	DAK-SYNAP	Sparsely positive

## Data Availability

The data presented in this study are available upon request from the corresponding author.

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
