# Peer review of "Extracutaneous Melanotic Melanoma with Nervous System Involvement in a Buffalo (Bubalus bubalis)"

_vetsci, 2023, doi:10.3390/vetsci10120662_

Round 1

Reviewer 1 Report

The review of paper entitled "Extracutaneous melanotic melanoma with nervous system involvement in a buffalo (Bubalus bubalis)"

Dear Authors,

You present a case report of a tumour in a buffalo that can be interesting to the readers. Nonetheless, in my opinion changes have to be made prior to publication.

My major concern is the quality of English. Due to multiple language flaws, the paper is very difficult to follow. I recommend an extensive English correction performed by either medically-experienced native speaker or medical English translator.

Other comments are as follow:

- in the Introduction section you mention that melanoma in cattle was reported only in two animals in Brazil, but as the neoplasms are usually geographically unrelated, it would be worth mentioning how many cases of cattle melanoma have been reported so far, regardless of localisation

- in the Case presentation, please include the sex, body weight and BCS of the animal; additionally, please reorganise the clinical description to make it clearer; please add the list of drugs used in the animal (including the doses and duration)

- the description of post-mortem examination would be easier to follow if the nervous system examination had been described after the necropsy description and before the statement that you found no cutaneous masses

- was the necropsy and/or histopathology performed by a board-certified pathologist?

- in the Case presentation you mention that the masses were found in the pre-scapular and sub-iliac lymph nodes (lines 64-65), while further in the text, you report neoplastic cells in "some mesenteric lymph nodes" (lines99-100) - please explain this inaccuracy

- please explain why the immunohistochemical analysis was performed only on the specimens from the brain?

- lines 101-104: were the antibodies tested for sensitivity and specificity in the buffalo species? If yes, the list of positive and negative controls should be attached; if the sensitivity and specificity was guaranteed by the antibodies' manufacturer, it should also be clearly stated (in Table 1 or as a Supplementary Material)

- in the Discussion section, it would be interesting to refer to other cases of melanomas in cattle reported in the literature

- in lines 151-152 you mention that SOX10 reactivity confirmed the neural crest origin of the neoplastic cells, while in Table 1 you report negative result for SOX10; moreover, please unify the SOX10 abbreviation throughout the manuscript, as at least three versions are being used by you: SOX10, SOX-10 and Sox-10

As mentioned before, the text requires a detailed correction performed by a professional medical English translator or native speaker.

There are many vocabulary grammatical mistakes that result in difficulties in following the text.

The quality of English is inappropriate for publication.

Reviewer 2 Report

The work is original, howevwer I removed some sentences in my opininion superfluous

The conclusions should be implemented and in my opinion it should also be specified whether radiological tests were carried out during the neurological deficit phases

minor editing of english language required

Reviewer 3 Report

The use of microscopic magnification of histological images should be considered: figure 2.

Consideration should be given to entering the dilution of the antibodies used in the table 1.

Line 136: add some bibliography to support "...as in many studies on melanomas in animals."

Line 46: Brazi(one l missing)

Line 64: In the necropsy, ....

Or The necropsy revealed tumor masses in the liver, ...

Line 99: ...adrenal gland, right kidney and in some...
